# A molecular epidemiological analysis of *Burkholderia pseudomallei* in southern Thailand

Jedsada Kaewrakmuk[1,2], Sarunyou Chusri[3], Pacharapong Khrongsee[4,5], Soontara Kawila[3], Vannarat Saechan[4], Nutjamee Leesahud[6], Bongkoch Chiewchanyont[6], Hathairat Thananchai[2], Kwanjit Duangsonk[2]*, Apichai Tuanyok[5]*

**1** Faculty of Medical Technology, Prince of Songkla University, Songkhla, Thailand, **2** Faculty of Medicine, Chiang Mai University, Chiang Mai, Thailand, **3** Faculty of Medicine, Prince of Songkla University, Songkhla. Thailand, **4** Faculty of Veterinary Science, Prince of Songkla University, Songkhla, Thailand, **5** Department of Infectious Diseases and Immunology, College of Veterinary Medicine, University of Florida, Gainesville, Florida, United States of America, **6** The Office of Disease Prevention Control 12, Songkhla, Thailand

\* kwanjit.d@cmu.ac.th (KD); tuanyok@ufl.edu (AT)

**Data Availability Statement:** All genome data are deposited under BioProject number: PRJNA1113702

## Abstract

Melioidosis, a severe bacterial illness caused by *Burkholderia pseudomallei*, is prevalent in most parts of Thailand, including its southern region situated within the Malay Peninsula. Despite a lower reported incidence rate of melioidosis in the South compared to the Northeast, the mortality rate remains persistently high. This study aimed to better understand the epidemiology and investigate the presence of *B. pseudomallei* in the natural environment of southern Thailand. Using multi-locus sequence typing (MLST), we characterized *B. pseudomallei* isolates derived from human cases and compared them with previously reported sequence types (STs) from the same region. A total of 263 clinical isolates retrieved from 156 melioidosis patients between 2014 and 2020 were analyzed, revealing 72 distinct STs, with 25 (35%) matching STs from Finkelstein's environmental isolates collected in southern Thailand during 1964–1967. Notably, strains bearing STs 288, 84, 54, 289, and 46 were frequently found among patients. Additionally, we observed strain diversity with multiple STs in 13 of 59 patients, indicating exposure to various *B. pseudomallei* genotypes in the environmental sources of the infection. Environmental surveys were conducted in Songkhla Province to detect *B. pseudomallei* in soil and water samples where local patients lived. Of the 2737 soil samples from 208 locations and 244 water samples from diverse sources, 52 (25%) soil sampling locations and 63 (26%) water sources were cultured positive for *B. pseudomallei*. Positive soil samples were predominantly found in animal farming area and non-agricultural zones like mountains and grasslands, while water samples were frequently positive in waterfalls, streams, and surface runoffs, with only 9% of rice paddies testing positive. Collectively, a significant proportion of recent melioidosis cases in Songkhla Province can be attributed to known *B. pseudomallei* STs persisting in the environment for at least the past six decades. Further characterization of *B. pseudomallei* isolates from recent environment surveys is warranted. These findings illuminate the contemporary landscape of *B. pseudomallei* infections and their environmental prevalence in southern Thailand, contributing to the regional threat assessment in Thailand and Southeast Asia.

**Funding:** This study was funded by the Faculty of Medicine, Chiang Mai University, Grant no. MC048-65 (Initials of authors receiving the grant: K.D.), and a startup fund from the Emerging Pathogens Institute under the preeminence professorship program at the University of Florida (Initials of authors receiving this fund: A.T). The funders had no role in study design, data collection and analysis, decision to publish, or preparation of the manuscript.

**Competing interests:** The authors have declared that no competing interests exist.

## Author summary

This study conducted a genetic analysis of *Burkholderia pseudomallei*, the bacterium causing a deadly tropical disease, from 156 patients in southern Thailand during 2014–2020. It found various genotypes, some of which were consistent with the genotypes found in the environment dating back to the 1960s. Extensive testing of soil and water in the region revealed contamination with *B. pseudomallei.* Areas such as animal farms, mountains, grasslands, and natural water sources e.g., streams and waterfalls, were mostly affected. This finding confirmed that some *B. pseudomallei* genotypes have had a long-standing presence in the environment, contributing to recent human cases. This study not only enhance our understanding of the spread of melioidosis throughout Southeast Asia but also highlights the importance of ongoing research and monitoring of *B. pseudomallei* in natural environments to address potential risks associated with the bacterium's persistence.

## Introduction

Melioidosis, a severe infectious disease caused by the Gram-negative bacterium *Burkholderia pseudomallei*, poses a substantial public health challenge in Thailand and throughout Southeast Asia—South Asia. This region is recognized as the global epicenter of melioidosis, with Thailand reporting one of the highest disease incidences worldwide [1,2]. Additionally, melioidosis is endemic in other tropical areas, including northern Australia, particularly in regions such as the Northern Territory, Queensland, and northern Western Australia [3–6]. The complex nature of melioidosis lies in its capacity to mimic various other illnesses, complicating accurate diagnosis [7]. Moreover, the bacterium's environmental resilience and antibiotic resistance further compound the problem [8]. Several factors contribute to the prevalence of melioidosis in Thailand, including the conducive tropical climate that supports the bacterium's survival in soil and water, as well as the high prevalence of risk factors such as diabetes and occupational exposure to contaminated environments from agricultural activities. Addressing the melioidosis challenge in Thailand necessitates improved surveillance, heightened awareness, and enhanced diagnostic capabilities to better manage this often underdiagnosed yet potentially fatal disease. Furthermore, a One Health approach and multidisciplinary research are paramount to mitigating the impact of melioidosis on both humans and animals in Thailand.

Although melioidosis is widespread in Thailand, its epidemiological understanding is limited, especially in lesser-studied regions like the North and the South. Since the initiation of the One Health initiative for melioidosis in southern Thailand by us in 2014, melioidosis cases have been reported in both humans and animals [9–11]. Notably, human cases predominantly manifest as bacteremia and pneumonia, with a substantial association with rainfall. Despite a lower incidence rate in these southern provinces compared to the highly endemic the northeastern regions, the mortality rate remains significantly high [9]. These recent findings underscore the importance of recognizing melioidosis in southern Thailand and emphasize the urgent need for enhanced disease surveillance and management within this region.

Previous genetic studies on *B. pseudomallei* have utilized both multi-locus sequence typing (MLST) and single nucleotide polymorphism (SNP) -based methods to understand the genetic diversity and epidemiology of this pathogen. MLST has been widely used to identify and compare sequence types (STs) across different regions, revealing significant genetic diversity within *B. pseudomallei* populations [12–17]. More recently, SNP-based analyses have provided

higher resolution insights into the genetic relationships and evolutionary history of *B. pseudomallei* strains, allowing for more detailed phylogenetic and population structure studies [16,17]. Environmental prevalence studies have shown that *B. pseudomallei* is commonly found in soil and water in various tropical regions, with significant implications for human exposure and infection risk [18,19]. Comparisons between clinical and environmental isolates have highlighted the importance of understanding the environmental reservoirs and transmission pathways of *B. pseudomallei* [20].

Historically, the presence of *B. pseudomallei* in the natural environment was investigated throughout Thailand, Malaysia, and Vietnam in a study led by Richard A. Finkelstein, a former SEATO (Southeast Asia Treaty Organization) microbiologist who was stationed in Bangkok from 1964 to 1967. The study, published in 2000, identified various environmental sources of *B. pseudomallei* in southern Thailand [21]. Most positive environmental samples were water samples collected from diverse sources. These findings laid the groundwork for understanding the environmental reservoirs of *B. pseudomallei* in the region.

In this current study, we aimed to extend this knowledge by characterizing *B. pseudomallei* isolates from southern Thailand and comparing them with environmental isolates in the historic Finkelstein collection using MLST to better understand the epidemiology and genetic diversity of the pathogen in this region. Additionally, we investigated specific genetic markers such as serogroups [22] and BTFC &YLF (*B. thailandensis*-like flagella and chemotaxis &*Yersinia*-like fimbrial gene clusters) genomic groups [23] to provide a more comprehensive understanding of the genetic diversity and distribution of *B. pseudomallei* in southern Thailand. These genomic groups, which have dissimilar geographic distributions, offer insights into the genetic and evolutionary differences among *B. pseudomallei* populations. Furthermore, environmental surveys were conducted to assess the presence of *B. pseudomallei* in the natural habitats of the affected patients.

## Materials and methods

### Ethics statement

This study was approved by the Office of Human Research Ethics Committee, Prince of Songkla University (Project no. REC 59-370-14-1), and from the Research Ethic Committee, Hatyai Hospital (Project no. REC-HY 52/2561). The *B. pseudomallei* isolated were collected routinely as part of the diagnostic procedures in the hospital laboratories. Formal consent from the patients was not obtained as the isolates were collected under the approved ethical protocols, which outlined that patient consent was not required due to the retrospective nature of the study and the use of anonymized data.

### Clinical *B. pseudomallei* isolates

*B. pseudomallei* isolates were collected from melioidosis cases identified by microbiology laboratories at Songklanakarind Hospital, Hatyai Hospital and Songkhla Provincial Hospital in Songkhla Province, Thailand, during 2014–2020 as a part of the One Health Initiative of southern Thailand melioidosis [9]. Of note, majority of these isolates were from cases identified during 2014 and 2017. Recent isolates from cases in 2018 and 2020 were also randomly selected. Multiple *B. pseudomallei* isolates from different diagnosed specimens were collected from each patient. The selection was based on the availability of isolates from different specimen types to provide a comprehensive understanding of the strain diversity in individual infections. In this study, 263 *B. pseudomallei* isolates from 156 patients were used (see S1 Table). All isolates were confirmed as *B. pseudomallei* by real-time PCR targeting TTS-1 gene marker as previously described [24].

### Environmental surveys for *B. pseudomallei*

Soil and water samples were collected from different locations within Songkhla Province. Sampling locations were chosen based on the places where local patients lived. These included residential areas and surrounding areas associated with agricultural activities, such as rice paddies, rubber plantations, fruit farms, and animal farms, and non-agricultural areas, such as mountains, grasslands, waterfalls, streams, and canals. The surveys were conducted in 75 out of 118 sub-districts within 15 out of 16 administrative districts of Songkhla Province during June 2019 to December 2022. We did not conduct the survey in Sabayoi District due to safety reasons regarding political conflict near Malaysia border. Geographic positioning system (GPS) coordinates of these sampling locations were recorded.

**a) Soil collection**: 2737 soil samples were collected from 208 locations in 15 districts. Each sampling site covered an area of approximately 100 m$^2$. Ten to 20 samples were collected from each site. A 2.5 meter-grid line was used to mark each digging hole. Two adjacent holes were 2.5 meters apart. The soil was dug by hand using a clean shovel. Twenty grams of soil was collected at 30 cm depth from the surface. Each sample was kept in a plastic zip bag and stored away from direct sunlight and high temperatures. To prevent contamination between samples, the equipment used for soil sampling was cleaned with tap water, rinsed with 70% alcohol, and dried by wiping up with paper towels between each sample collection. Soil types based on their textures were described and recorded. Soil colors were measured using Munsell Soil Color Charts (Pantone LLC., New Jersey, USA). Soil pH were measured from soil solution made in the following step using pH test strips. The soil samples were processed on the same day they were collected.

**b) Water collection**: 244 water samples were collected from various sources in Songkhla Province. These included domestic boreholes, canals, mangrove forests or wetlands, reservoirs, stagnant water in rice paddies, streams, springs or waterfalls, surface or overland runoffs, and private or community water wells. One sample was collected at each location by thoroughly filling a 1-liter sterile plastic bottle under the surface of the water source. Physical data, such as pH, color, and turbidity, were recorded. The samples were processed on the same day.

**Isolation of *B. pseudomallei* from soil samples.** Upon arriving in the laboratory, each soil sample was mixed thoroughly in the collecting bag by shaking with hands and then weighted to aliquot into two equal portions of 10 g each in a sterile 50 mL conical tube for use in the following steps.

**a) Direct culture**: 10 g of soil sample contained in a sterile 50 mL conical tube was added with 10 mL of sterile distilled water. The tube was vortexed to allow the soil and water to mix thoroughly. The tube was let to stand at room temperature overnight. The next day, 100 μL of soil solution was cultured by streaking onto an Ashdown's agar plate [25]. The plate was incubated at 37˚C for seven days. Bacterial colony formations were observed daily. Suspected *B. pseudomallei* colonies were subjected to testing with a latex agglutination test [3].

**b) Enrichment culture**: 10 g of soil sample contained in a sterile 50 mL conical tube was added with 10 mL of enrichment broth (Tryptic soy broth supplemented with 0.1% (w/v) crystal violet, 4 mg/L gentamicin). The content was mixed thoroughly by vortexing, and the tube was incubated at 42˚C without shaking overnight. On the next day, two volumes, 10 and 100 μL of enriched cultures, were inoculated by streaking onto an Ashdown's agar plate. The plates were incubated at 37˚C for seven days. Bacterial colony formations were observed daily. Suspected *B. pseudomallei* colonies were tested using a latex agglutination test [3].

Both the direct and the enrichment soil cultures described above followed the consensus guidelines [26] with the exception that gentamicin was used instead of colistin in the enrichment broth.

**Isolation of *B. pseudomallei* from water samples.** Three culturing techniques were used and summarized in a flow diagram (Fig 1).

**Colony PCR.** All positive bacterial colonies identified by latex agglutination test were further tested by real-time PCR. *B. pseudomallei* -specific TTS-1 assay was used as previously described [24]. A tiny piece of a bacterial colony was picked up by a pipet tip and then added

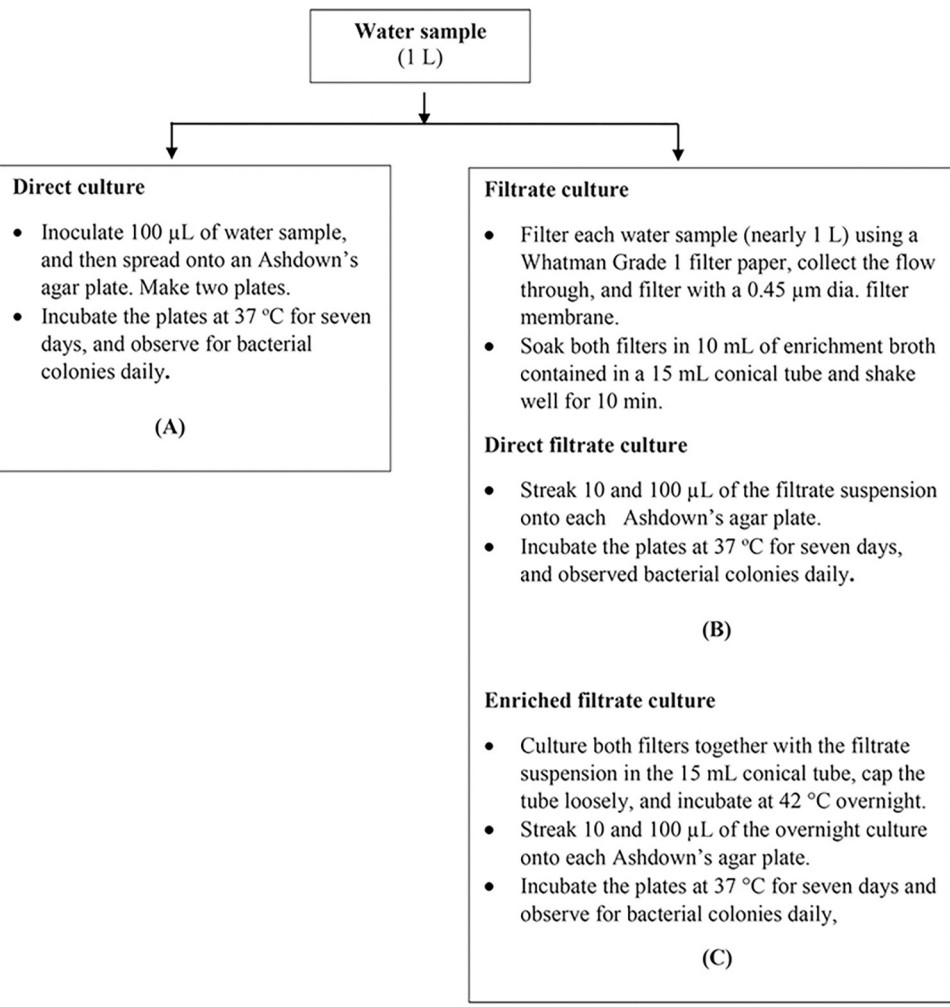

**Fig 1. The investigative flow of three different culturing methods applied to water samples:** (A) direct culture, (B) direct filtrate culture, and (C) enriched filtrate culture. A: Direct culture: 100 μL of water sample was inoculated and spread onto an Ashdown's agar plate. Two plates were used for each water sample. The plates were incubated at 37˚C for seven days. Bacterial colony formations were observed daily. Suspected *B. pseudomallei* colonies were tested using a latex agglutination test. Filtrate culture: The rest of the water sample, nearly 1 L, was prefiltered with a Whatman Grade 1 filter paper using a sterile grass funnel, and the flow through was collected in a sterile Erlenmeyer flask. The filtered paper was picked up and kept in a 15 mL sterile conical tube. The flow through was then further filtered with a 0.45 μm dia. - 47 mm filter membrane (mdi Membrane Technology, PA, USA) using a vacuum filtration system. The filtered membrane was picked up and kept together in the same 15 mL conical tube. Then, 10 mL of the enrichment broth was added to the tube. The tube was shaken at 250 rpm on a shaker at room temperature for 10 min. Two culturing techniques were conducted as follows: B: Direct filtrate culture: Two volumes, 10 and 100 μL, of the filtrate solution were inoculated by streaking onto each Ashdown's agar plate. The plates were incubated at 37˚C for seven days. Bacterial colony formations were observed daily. Suspected *B. pseudomallei* colonies were tested using a latex agglutination test. C: Enriched filtrate culture: The filtrate solution together with both filters contained in the 15 mL conical tube was incubated at 42˚C overnight. On the next day, two volumes, 10 and 100 μL of enriched cultures, were inoculated by streaking onto an Ashdown's agar plate. The plates were incubated at 37˚C for seven days. Bacterial colony formations were observed daily. Suspected *B. pseudomallei* colonies were tested using a latex agglutination test.

to the PCR master mix instead of using a purified DNA sample as usual. The PCR assay was run in a portable dual-probe real-time PCR machine, CHAI Open qPCR (Santa Clara, CA, USA). PCR results were considered positive if their Ct values were less than 30.

**Multi-locus sequence typing (MLST).**   MLST analysis was conducted using two different techniques: 1) PCR amplicon-based sequencing of 7 housekeeping genes: *ace*, *gltB*, *gmhD*, *lepA*, *lipA*, *nark*, and *ndh*, as previously described [27], analyzed in all 263 isolates, or 2) *in silico* analysis of the whole genome sequencing data. Genomic DNA of *B. pseudomallei* isolates extracted using Wizard Genomic DNA Purification Kit (Promega Inc., Wisconsin, USA) were used in both techniques. The latter, the DNA samples from 126 selected isolates were sequenced using Illumina sequencing methods. Briefly, Illumina sequencing libraries were prepared using the tagmentation-based and PCR-based Illumina DNA Prep kit and custom IDT 10 bp unique dual indices (UDI) with a target insert size of 320 bp. No additional DNA fragmentation or size selection steps were performed. Illumina sequencing was performed on an Illumina NovaSeq 6000 sequencer in multiplexed shared-flow-cell runs, producing 2x151 paired-end reads. Demultiplexing quality control and adapter trimming was performed with bcl-convert version v4.1.5. (Illumina Inc., San Diego, California, USA). The sequencing data of 126 genomes of *B. pseudomallei* sequenced in this study are available as the GenBank's Sequence Read Archive (SRA) under the BioProject accession number PRJNA1113702. Genomic assemblies were performed using programs on Galaxy, a web-based framework for biological computing at the University of Florida. Briefly, genome contigs were generated from the FASTQ files using a Galaxy workflow: FASTQ Groomer (Galaxy Version 1.1.1) and SPAdes genome assembler (Galaxy Version 3.12.0+galaxy1) with default kmers: 21, 35, and 55. Each genome contained 200–300 contigs with approximately 30-50X coverage. The local BLASTN function on BioEdit version 7.0.5.3 was used to search for MLST gene allele sequences [28]. The allele sequences of seven housekeeping genes including, *ace*, *gltB*, *gmhD*, *lepA*, *lipA*, *nark*, and *ndh* of *B. pseudomallei* K96243 (Sequence Type 10), were used as the query sequences. All seven housekeeping gene sequences extracted from these genome data were used in the analysis. New gene allele numbers and new sequence types were assigned by pubMLST (https:// pubmlst.org).

**Phylogenetic analysis.**   Sequence Type—based genetic relatedness analysis were performed using Global optimal eBURST (goeBURST) distance–minimal spanning tree algorithm [29] to display STs' similarity and variance was displayed on PHYLOViZ 2.0 [30], a platform-independent Java program that allowed phylogenetic inference and data visualization for large datasets of sequence based typing method.

**GPS mapping.**   The recorded GPS coordinates of the environmental sampling sites were located on a map of Songkhla Province using QGIS Desktop 3.16.9 software (Stack Exchange Inc., New York, USA). The map was obtained from the Southern Regional Center of Geo-Informatics and Space Technology, Prince of Songkla University, Hatyai, Songkhla, Thailand.

## Results

### Genotypic characterization of *B. pseudomallei* isolates from patients

**Sequence types (STs).**   There were 72 different sequence types (STs) identified from 263 clinical *B. pseudomallei* isolates by MLST. These isolates were collected from 156 patients admitted for treatments at three tertiary-care hospitals in Songkhla Province. Strains with ST288, ST84, ST46, ST54, and ST289 were the five most common STs which were found in 24, 13, 11, 8, 8, patients, respectively. There were 22 novel STs, including ST1382, ST1383, ST1384, ST1385, ST1386, ST1387, ST1388, ST1389, ST1390, ST1391, ST1466, ST1467, ST1469, ST1470, ST1471, ST1522, ST1523, ST1562, ST1645, ST2059, ST2060, and ST2061, which were

found in one or two patients. To study the strain diversity in individual infection, at least two *B. pseudomallei* isolates were collected from cultures of available clinical specimens (e.g. blood, sputum, pus, urine) in 59 patients. We noted that as many as 11 isolates were collected from one patient (Patient ID# p038, see S1 Table). MLST has revealed that *B. pseudomallei* isolates from 13 out of these 59 (22%) patients mostly had two STs, except one patient (#p209) had three distinct STs. Details of all 263 *B. pseudomallei* isolates used in this study and their STs are shown in S1 Table.

**Genomic groups.** Two distinct genomic groups within *B. pseudomallei* that have dissimilar geographic distribution, known as BTFC (*B. thailandensis*-like flagellum and chemotaxis) gene cluster and YLF (Yersinia-like fimbrial) gene cluster, were identified in all tested *B. pseudomallei* isolates by PCR as previously described [23]. As expected, most isolates belonged to YLF group, except the isolates from two patients, including a local patient #p206 from Songkhla Province and a referred patient #p231 from Phuket Province, were placed into the BTFC group. The isolates from both patients also had different STs, ST834 and ST39, respectively.

**Serogroups.** LPS typing was used to identify the serogroup of all *B. pseudomallei* isolates by PCR as previously described [22]. Most *B. pseudomallei* isolates had LPS type A, except one isolate recovered from a multiple-strain infection in patient #p063 had LPS type B. This LPS type B isolate was identified as ST314, while another isolate from the same patient had ST306 with LPS type A.

**Isolation of *B. pseudomallei* from soil.** Out of the total of 2737 soil samples collected from 208 sampling locations, 116 (4%) samples from 52 (25%) sampling locations yielded the presence of *B. pseudomallei* through culturing. On average, two soil samples from each of these positive sampling locations were tested positive for *B. pseudomallei*. These soil samples were classified into eleven groups based on their sources, as detailed in Table 1. The highest incidence of positive samples was found in non-agricultural areas, such as hills, mountains, springs, or waterfalls, with 61% of the sampling locations (14 out of 23) exhibiting the presence of *B. pseudomallei*. Of the agricultural areas, soil samples from animal farms were most positive. Four (40%) out of 10 animal farms were positive. We also noted that soil samples from rice paddies were less positive. Only 9% (3 out of 35) of the rice paddies, mostly collected from

**Table 1. The number of soil samples collected from different sources positive for *B. pseudomallei* by culturing technique.**

| Soil sources | No. of positive location (%) | No. of positive sample (%) |
|---|---|---|
| Animal farms | 4/10 (40%) | 22/163 (13%) |
| Canal/reservoir/stream bank* | 17/65 (26%) | 30/705 (4%) |
| Crop farms (cassava/ corn/ sugarcane/ vegetable) | 2/6 (33%) | 3/85 (4%) |
| Coconut plantation/ Oil palm plantation/ Fruit farms/ Rubber plantation | 4/31 (13%) | 9/420 (2%) |
| Grassland | 7/27 (26%) | 17/405 (4%) |
| Hill/ Mountain/ Spring/ Water fall* | 14/23 (61%) | 29/329 (9%) |
| Mangrove Forest * | 0/2 (0%) | 0/20 (0%) |
| Residential area | 0/7 (0%) | 0/130 (0%) |
| Rice paddies | 3/35 (9%) | 4/290 (1%) |
| Offshore islands* | 1/2 (50%) | 2/60 (3%) |
| Total | 52/208 (25%) | 116/2737 (4%) |

*undisturbed, non-agricultural area

Singha Nakorn District, were positive for the presence of *B. pseudomallei*. In addition, none of the samples from seven residential areas were tested positive.

## Isolation of *B. pseudomallei* from water

Out of a total of 244 examined water samples from various sources or locations, 63 samples (26%) were tested positive for the presence of *B. pseudomallei* through at least one culturing method. Among these 63 positive samples, direct cultures detected 10 (15.9%) of them, filtrate cultures identified 35 (55.6%), and enriched filtrate cultures found 54 (85.7%) samples. Additionally, 24 positive samples (38.1%) were by more than one culturing method. These positive water samples were collected from various sources, including surface or overland runoffs (88%), springs or waterfalls (49%), streams (23%), reservoirs (21%), canals (18%), and private or community water wells (6%). It is worth noting that no *B. pseudomallei* was detected in water samples collected from domestic boreholes, lakes in mangrove forests or wetlands, and stagnant water in rice paddies. Most of the positive samples were associated with natural water sources, as detailed in Table 2. A full list of water samples, sourced locations and their culturing results is shown in S2 Table.

**Physical properties of soil and water.** The soil samples collected displayed a diverse range of types, characterized by a wide pH scale from 4.0 to 8.0. These soils featured a variety of colors, ranging from dark gray (matching the Munsell soil color code of 10YR2/1) to a vivid red (7.5YR7/8). The textures of these soils also varied significantly, encompassing everything from clay to laterite, a type of reddish and clay-rich soil. Soil samples testing positive for *B. pseudomallei* were found across most soil types, including clayey soil (color code 10YR2/1) and laterite (color code 7YR/7/6), with an average pH value of 6.0. Notably, no *B. pseudomallei* positive samples were found in soils with a pH of 8.0. The collected water samples, all of which were fresh water, demonstrated a varied pH range from 4.5 to 7.0. Among the 56 samples that cultured positive, only seven were muddy water, whereas the rest were clear and contained minimal sediment. The pH levels of these positive samples spanned from 4.5 to 7.0.

**Geographic distribution of *B. pseudomallei* in the environment.** Geographic locations of the *B. pseudomallei* tested positive or negative environmental samples were mapped based on their GPS coordinates. These locations are shown in Fig 2, and the numbers of positive samples and locations are summarized in Table 3. We noted that *B. pseudomallei* was detected in the environmental samples collected 11 out of 15 districts. Soil and water samples from four

**Table 2. The number of water samples collected from different sources positive for *B. pseudomallei* by culturing technique.**

| Water source | No. of positive locations (%) |
| --- | --- |
| Domestic boreholes | 0/8 (0%) |
| Canals* | 4/22 (18%) |
| Lakes in mangrove forests or wetlands* | 0/6 (0%) |
| Reservoirs* | 6/28 (21%) |
| Water in rice paddies | 0/13 (0%) |
| Streams (streamlets, brooks or creeks) * | 22/95 (23%) |
| Springs or waterfalls* | 23/47 (49%) |
| Surface or overland runoffs* | 7/8 (88%) |
| Private/community water wells | 1/16 (6%) |
| Total | 63/244 (26%) |

*non-agricultural areas

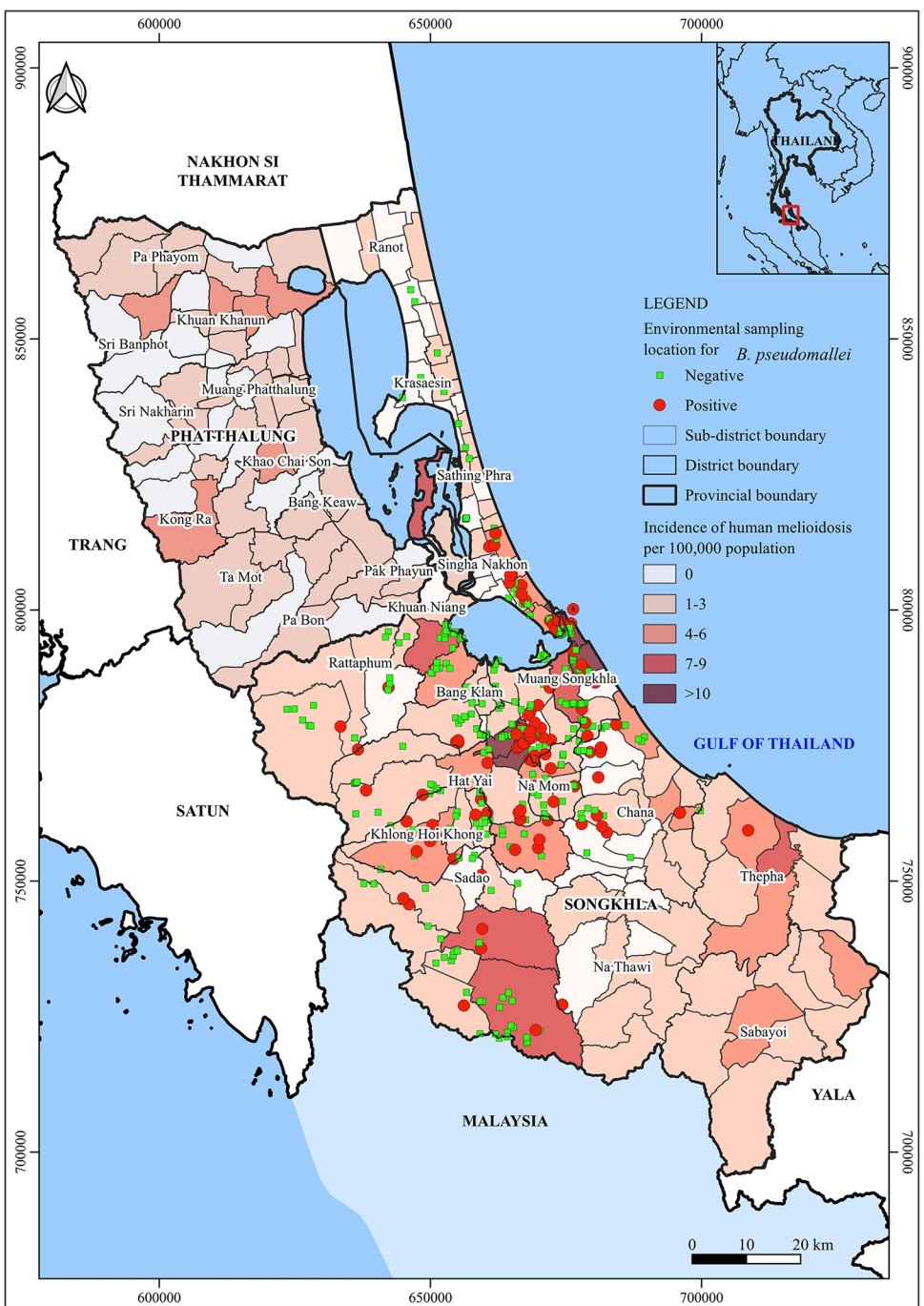

**Fig 2. GIS mapping of *B. pseudomallei* distribution in Songkhla Province, Southern Thailand.** The map employs a color-gradient to represent the average incidence rate of melioidosis in each subdistrict of Songkhla Province and Phatthalung Province, a nearby province, as previously reported. The environmental surveys spanned 15 different districts. Locations where *B. pseudomallei* was confirmed in soil or water samples are marked with red dots. Conversely, green squares denote sampling sites where cultures tested negative for *B. pseudomallei*. Note: The base map shape file was authentically created in-house by the Southern Regional Center of Geo-Informatics and Space Technology Prince of Songkla University for use in this study. The recorded GPS coordinates of the environmental sampling sites were located on the map using QGIS Desktop 3.16.9 software (Stack Exchange Inc., New York, USA).

**Table 3. The occurrence of *B. pseudomallei* in the environments surveyed in 15 out of 16 administrative districts of Songkhla Province.**

| District | Known incidence rate/100,000 population | Positive sampling sites (%) | No. of soil samples tested positive | No. of water samples tested positive |
|---|---|---|---|---|
| Krasaesin | 0.00 | 0/2 (0.0%) | 0/20 | 0/2 |
| Rattaphum | 0.95 | 3/17 (17.6%) | 2/145 | 2/6 |
| Ranot | 1.33 | 0/4 (0.0%) | 0/40 | 0/4 |
| Na Thawi | 1.85 | 1/2 (50.0%) | 2/30 | 1/2 |
| Hatyai | 1.99 | 34/87 (39.1%) | 23/442 | 26/62 |
| Muang Songkhla | 3.05 | 11/54 (20.4%) | 7/360 | 5/32 |
| Sadao | 3.10 | 7/44 (15.9%) | 13/353 | 0/27 |
| Bang Klam | 3.12 | 3/27 (11.1%) | 7/165 | 0/17 |
| Chana | 3.17 | 11/25 (44.0%) | 9/135 | 7/16 |
| Sathing Phra | 3.57 | 0/11 (0.0%) | 0/60 | 0/6 |
| Sabayoi | 3.94 | No data | No data | No data |
| Thepa | 4.54 | 1/2 (50%) | 1/40 | 0/2 |
| Singha Nakhon | 4.59 | 17/41 (41.5%) | 16/385 | 9/19 |
| Kuan Niang | 5.00 | 0/20 (0.0%) | 0/155 | 0/12 |
| Klong Hoi Khong | 5.26 | 9/25 (36.0%) | 27/255 | 5/11 |
| Namom | 6.23 | 11/33 (33.3%) | 9/152 | 8/26 |

districts, including Krasaesin, Ranot, Sathing Phra, and Kuan Niang, were tested negative for *B. pseudomallei*. Of note, these districts are known as the rice farming districts of Songkhla Province where most soil and water samples were collected from rice paddies for analysis.

## Discussion

This study aimed to characterize *B. pseudomallei* isolates collected from melioidosis patients who were admitted to three tertiary care hospitals in Songkhla Province of southern Thailand during 2014–2020. The strains' sequence types (STs) were compared with those that have been discovered in the natural environment of southern Thailand. The presence of *B. pseudomallei* in the natural environment was investigated through a comprehensive survey of soil and water collected from different sources in Songkhla Province. Here are the major points to discuss:

*Recent melioidosis cases were caused by known strains' STs found to be present in the natural environment of southern Thailand, dating back to the 1960s.* In the historic study by Finkelstein and colleagues, most positive environmental samples were water samples collected from various sources in southern Thailand [21]. *B. pseudomallei* isolates were collected from heart blood cultures of sick or deceased laboratory hamsters injected intraperitoneally with water samples or soil solutions. These isolates were further genotyped in a follow-up study by McCombie and colleagues almost 40 years later [13]. Of the 207 isolates, 143 were from freshwater sources throughout southern Thailand. Out of 80 different STs identified, 61 were from this region. In our study, 25 (34.7%) of 72 STs in patients identified matched those from Finkelstein's collection, including the five most common STs (288, 84, 54, 289, and 46).

In another study investigating an outbreak in Koh Phangan, an island in the Gulf of Thailand, in 2012. Twelve different STs were identified, with two (ST164 and ST385) found in three human cases, and ten, including seven novel STs, identified from water sources [31]. Their finding could suggest a significant degree of diversity in the *B. pseudomallei* population even present on a small island of approximately 48.3 square miles in southern Thailand. ST164 and ST385 were also found in our patients, though none of the water supply STs matched the

patient isolates. Additionally, we reported *B. pseudomallei* isolate with ST366 from a canine melioidosis case in Pattani Province [10], and ST392 from a swine case linked to in a bore-hole water supply used on a pig farm, along with three other STs (ST51, ST298, and ST1721) from soil on pig farms in Nakhon Si Thammarat Province [11]. ST366 matched recent human and historic environmental isolates, while ST392 only matched two historic environmental strains, but not human cases.

To date, 119 different STs, including those from our study, have been identified in *B. pseudomallei* strains from southern Thailand. A combined list of these STs is shown in Table 4. Furthermore, goeBURST analysis demonstrated genetic distance differences among these STs, identifying two major groups separated by a double-locus variant (DLV) distance between ST507 and ST164, (Fig 3).

In Northeast Thailand where melioidosis is highly endemic, up to seven different STs have been identified from a single soil sample [16]. Early MLST studies, such as by Vesaratchavest and colleagues compared STs between soil and patient isolates from Northeast Thailand, revealing significant genotypic differences between the groups [12]. This suggests non-random distribution of *B. pseudomallei* genotypes. Similarly, in our study, the five most common STs were found in up to 41.0% of patients, indicating potential overrepresentation. Notably, 34.7% of the STs in patients isolates matched those in Finkelstein's environmental strains, likely because Finkelstein's strains were selected from STs causing disease in hamsters. Therefore, the 60 different STs reported from southern Thailand in Finkelstein's study likely represent only invasive strains, and the true diversity of *B. pseudomallei* in the natural environment remains unknown.

***Multiple STs associated with individual's infection could demonstrate a certain level of strain diversity within the environmental source.*** In this study, we genotyped multiple *B. pseudomallei* isolates, predominantly from hemocultures, collected from 59 selected patients. The purpose was to analyze the diversity of strains responsible for individual infections since strains' diversity in the environmental source of the infection could be high. Notably, 13 of these patients, accounting for 22%, were infected with 2 to 3 different STs. Of note, one patient was co-infected by both serotype A and B strains. According to the literature review, most existing studies typically genotyped only one or two isolates from each patient [15,32–34]. This approach often fails to reveal any genotypic differences during the initial infection episode. It has been discussed upon subsequent episodes of melioidosis that if the same genotype is present, the case can be classified as a relapse or a recurrent infection [35,36]. Conversely, if a different genotype emerges in a subsequent episode which may take months or years later, it is then considered a reinfection. Using a higher resolution genotyping technique, such as whole genome SNP analysis, would be more effective in differentiating between relapse and infection. Given that certain STs have been present in the environment for a long time, a comprehensive comparative genomic analysis between clinical and environmental strains sharing the same ST would provide valuable insights into genetic changes, such as gene acquisition or loss, over time. Therefore, we suggest genotyping multiple isolates from the initial infection episode in geographic areas where *B. pseudomallei* is highly diverse, such as Thailand. Furthermore, a recent study has shown that not all soil isolates even though they were isolated from the same soil sample caused infection in BALB/c mice via subcutaneous infection [11]. Specifically, only 30 out of 50 soil isolates killed mice suggesting the variable virulence potential of different *B. pseudomallei* isolates could be present in the same environmental source of infection.

***B. pseudomallei isolates from southern Thailand exhibited several STs that were identical to those found in strains from Peninsular Malaysia.*** A recent comprehensive study conducted in Malaysia identified 29 distinct STs from 84 *B. pseudomallei* isolates, collected from septicemic patients across 14 states in both Peninsular and Borneo Malaysia [14]. Interestingly, 16

**Table 4. A combined list of the sequence types (STs) of *B. pseudomallei* isolates identified in human cases in this study, the environmental strains previously isolated from southern Thailand in the historic Finkelstein's strain collection, and strains from other studies conducted in southern Thailand.**

| Sequence type (ST) | No. of patients (this study) | Environmental isolates (Finkelstein's) | Strains presented in other reports from southern Thailand | MLST database | |
|---|---|---|---|---|---|
| | | | | Source | Country |
| 3 | 2 | STW 487–1 | | H, E | Thailand |
| 10 | 1 | Phangna 64 W | | H, E | Thailand, Malaysia |
| 15 | 2 | STW 100–1 | | H, E | Thailand, China, Cambodia |
| 38 | 0 | Ranong 70 W | | H, E | Thailand |
| 39 | 1 | - | | H | Thailand |
| 46 | 11 | Phuket 6 S-1, STW 22–1, STW 25, STW 185 | | H, A, E | Thailand, Malaysia, Singapore, Vietnam, Bangladesh, Indonesia, Australia, New Zealand (ex. Thailand), China, Cambodia |
| 50 | 1 | - | | H, E | Thailand, Malaysia, Singapore China, |
| 51 | 2 | STW 98–1, STW 104–1, STW 110–1, STW 115–2, STW 181–1, STW 205–1, STW 208, STW 208, STW 430 | Soil from a pig farm | H, E | Thailand, Malaysia, Singapore, China |
| 54 | 8 | Phuket 3 W-1, STW 106–1, STW 168–3 | | H, E | Thailand, Malaysia, Singapore, Cambodia, Indonesia, Laos, UK (ex. Asia) |
| 56 | 1 | - | | H, E | Thailand, Malaysia, Bangladesh, Cambodia, Vietnam, Burma |
| 70 | 2 | STW 27–2, STW 765 | | H, A, E | Thailand, Laos, China, Indonesia, Australia, Vietnam |
| 84 | 13 | Songkhla 34 W-2, STW 38, STW 62, STW 97–1, STW 101–1, STW 102–3, STW 214, STW 225–3 | | H, A, E | Thailand, Malaysia, Singapore, Australia |
| 93 | 3 | - | | H, E | Thailand |
| 99 | 1 | - | | H, E | Thailand, Cambodia, Philippines, Malaysia |
| 153 | 1 | - | | H | Thailand |
| 164 | 3 | Phattalung 52 W-2, STW 94–1, STW 111–2, STW 406 | Koh Phangan—case #2 and #3 | H, E | Thailand, Malaysia |
| 168 | 3 | Songkhla 11 W, STW 122, STW 157 | | H, E | Thailand, Malaysia |
| 174 | 1 | - | | H, E | Thailand, China, Australia |
| 205 | 1 | - | | H, E | Thailand, Malaysia, China |
| 206 | 2 | - | | H, E | Thailand, Cambodia |
| 227 | 0 | Ranong 73 W-2 | | H, E | Thailand |
| 228 | 0 | Songkhla 21 W-2 | | H, E | Thailand, Vietnam |
| 271 | 1 | - | | H, A | Malaysia, China |
| 288 | 24 | STW11-3 | | H, E | Thailand, Malaysia, Vietnam |
| 289 | 8 | Phattalung 49 W-2, Songkhla 21 W-1, STW 5–1, STW 45–1, STW 55–2, STW 58–1, STW 116–2, STW 162–1, STW 220–2, STW 224–1 | | H, A, E | Thailand, Malaysia, Singapore |
| 290 | 1 | STW 415 | | H, E | Thailand |
| 298 | 0 | - | Soil from a pig farm | H | Thailand |
| 300 | 1 | STW 305, STW 312–1 | | H, E | Thailand, Bangladesh, India, Burma |
| 306 | 1 | - | | H | Thailand, Malaysia, China |
| 312 | 0 | STW 358–2 | | H, E | Thailand |
| 314 | 3 | - | | H | Thailand |
| 364 | 0 | STW 359–1, STW 362, STW 364, STW 368–3 | | E | Thailand |
| 366 | 4 | STW 3, STW 4–2, STW 36–1, STW 186–2, STW 187–3, STW 202–3, STW 204–2, STW 230–1, STW 447 | A canine case | H, E | Thailand, Malaysia, China, Vietnam |
| 368 | 2 | STW 219 | | H, E | Thailand, Malaysia, Laos |
| 369 | 2 | Phattalung 49 w-1, Phattalung 51 W, Songkhla 27 W-1, Songkhla 27 W-2, STW 1, STW 114–1, STW 175–1 | | H, E | Thailand, Malaysia, Vietnam |

*(Continued)*

**Table 4.** (Continued)

| Sequence type (ST) | No. of patients (this study) | Environmental isolates (Finkelstein's) | Strains presented in other reports from southern Thailand | MLST database | |
|---|---|---|---|---|---|
| | | | | Source | Country |
| 371 | 1 | STW 35–1, STW 39, STW 42, STW 43–1, STW 44, STW 95–1, STW 96–2, STW 105–1, STW 117–4, STW 154, STW 221–1, STW 233–1 | | H, E | Thailand, Malaysia |
| 372 | 0 | STW 61–2, STW 107–1 | | E | Thailand |
| 374 | 0 | STW 67–1 | | E | Thailand |
| 376 | 3 | STW 7, STW 7–2, STW 28–5, STW 197–1, STW 199–2, STW 200–1, STW 235–1, STW 760–2 | | H, E | Thailand, Malaysia, Laos, Australia, Portugal (ex. Thailand) |
| 377 | 0 | Chumphon 76 W-2 | | E | Thailand |
| 378 | 0 | STW 222–2, STW 244–1 | | E | Thailand |
| 379 | 0 | STW 402 | | E | Thailand |
| 380 | 0 | STW99-2 | | E | Thailand |
| 381 | 0 | STW 640 | | H, E | Thailand, Cambodia, Vietnam |
| 382 | 0 | Songkhla 25 W-1, Songkhla 25 W-2 | | E | Thailand |
| 383 | 0 | STW 217–2 | | E | Thailand |
| 384 | 1 | STW 66 | | E | Thailand |
| 385 | 3 | STW 26, STW 33, STW 64, STW 551 | Koh Phangan–case #1 (a newborn) | E | Thailand |
| 386 | 0 | STW 754 | | H, E | Thailand |
| 388 | 0 | STW 723 | | E | Thailand |
| 389 | 0 | STW 32–1, STW 120–1 | | H, E | Thailand, Vietnam |
| 390 | 0 | STW 414–1 | | E | Thailand |
| 391 | 0 | STW 422 | | E | Thailand |
| 392 | 0 | STW 10, STW 539–1 | A swine case, and contaminated water supply | H, E | Thailand |
| 393 | 0 | STW 561–1 | | H, E | Thailand, Cambodia, Australia (ex. Thailand) |
| 394 | 0 | STW 729 | | E | Thailand |
| 395 | 3 | STW 307–2 | | E | Thailand |
| 396 | 0 | STW 189–2 | | E | Thailand |
| 398 | 0 | STW 638–1 | | E | Thailand |
| 399 | 0 | Ranong 73 W-1 | | E | Thailand |
| 400 | 0 | STW 216–2 | | E | Thailand |
| 401 | 0 | STW 176 | | E | Thailand |
| 402 | 2 | STW 174 | | H, E | Thailand, Malaysia |
| 404 | 1 | STW 730–1 | | H, E | Thailand, Japan (travel Thailand) |
| 405 | 0 | STW 185–1 | | H, E | Thailand, China, India |
| 407 | 0 | STW 32–3, STW 34 | | E | Thailand |
| 408 | 0 | STW 215–1 | | E | Thailand |
| 409 | 0 | Ranong 8, STW 426–2 | | E | Thailand |
| 411 | 1 | - | | H | Thailand, Vietnam |
| 412 | 0 | STW 753 | | E | Thailand |
| 413 | 0 | STW 429 | | E | Thailand |
| 414 | 3 | Songkhla 34 W-1 | | H, E | Thailand, Malaysia, Singapore, France (ex. Malaysia) |
| 415 | 0 | STW 28–2 | | E | Thailand |
| 416 | 0 | STW 152 | Koh Phangan—water | E | Thailand |
| 417 | 0 | STW 420–2, STW 424–1 | | E | Thailand |

*(Continued)*

**Table 4.** (Continued)

| Sequence type (ST) | No. of patients (this study) | Environmental isolates (Finkelstein's) | Strains presented in other reports from southern Thailand | MLST database | |
|---|---|---|---|---|---|
| | | | | Source | Country |
| 423 | 1 | - | | H, E | Singapore, Malaysia |
| 507 | 2 | - | | H, E | Malaysia, Laos |
| 594 | 4 | - | | H, E | Australia, Australia, Sri Lanka |
| 653 | 1 | - | | H | Thailand, Cambodia |
| 658 | 1 | - | | H | Thailand, Malaysia, Laos, China |
| 707 | 0 | - | Koh Phangan-water | H | UK (ex. Nigeria) |
| 834 | 2 | - | | H | Cambodia |
| 930 | 0 | | Koh Phangan—water | H, E | Cambodia, Australia, China, Nigeria |
| 1057 | 1 | - | | H, E | Thailand, Malaysia, Singapore |
| 1113 | 0 | - | Koh Phangan—water | E | Thailand |
| 1114 | 0 | - | Koh Phangan—water | E | Thailand |
| 1115 | 0 | - | Koh Phangan—water | E | Thailand |
| 1116 | 0 | - | Koh Phangan—water | E | Thailand |
| 1117 | 0 | - | Koh Phangan—water | E | Thailand |
| 1118 | 0 | - | Koh Phangan—water | E | Thailand |
| 1119 | 0 | - | Koh Phangan—water | E | Thailand |
| 1323 | 3 | - | | H | Thailand, Malaysia |
| 1359 | 3 | - | | H | Thailand, Malaysia, Cambodia |
| 1364 | 1 | - | | H | Sri Lanka |
| 1382 | 1 | - | | | This study |
| 1383 | 1 | - | | | This study |
| 1384 | 1 | - | | | This study |
| 1385 | 1 | - | | | This study |
| 1386 | 1 | - | | | This study |
| 1387 | 1 | - | | | This study |
| 1388 | 1 | - | | | This study |
| 1389 | 1 | - | | | This study |
| 1390 | 1 | - | | | This study |
| 1391 | 1 | - | | | This study |
| 1466 | 1 | - | | | This study |
| 1467 | 1 | - | | | This study |
| 1468 | 1 | - | | H | Malaysia |
| 1469 | 2 | - | | | This study |
| 1470 | 1 | - | | | This study |
| 1471 | 1 | - | | | This study |
| 1522 | 2 | - | | | This study |
| 1523 | 2 | - | | | This study |
| 1562 | 1 | - | | | This study |
| 1645 | 1 | - | | | This study |
| 1721 | 0 | - | Soil from a pig farm | E | Thailand |
| 1736 | 1 | - | | H | Malaysia |
| 2059 | 1 | - | | | This study |
| 2060 | 1 | - | | | This study |
| 2061 | 1 | - | | | This study |

Note: H, human; A, animal; E, environment

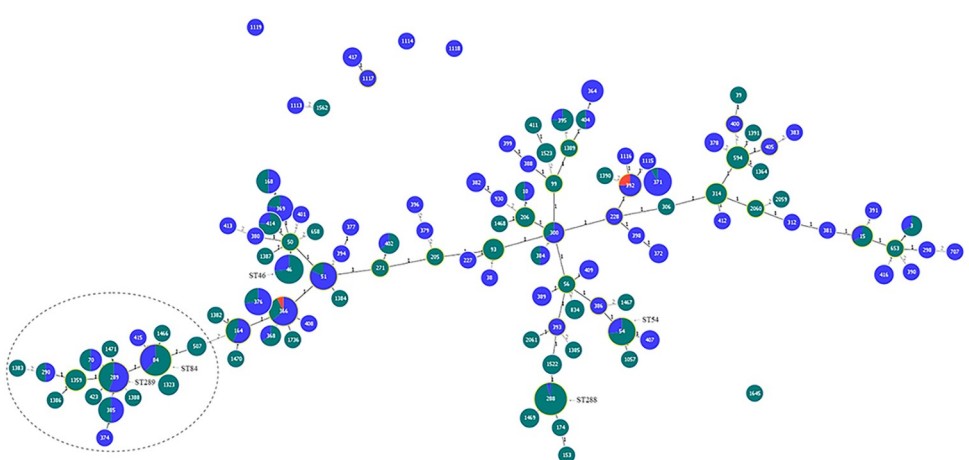

**Fig 3. goeBURST analysis of sequence type (ST) diversity of *B. pseudomallei* identified in patients' isolates in this study, and those found in the environment and animal cases in southern Thailand.** So far, a total of 119 different STs of *B. pseudomallei* were reported from southern Thailand, at least one-third of the STs identified in human melioidosis cases had known STs ever reported in the Finkelstein's historic environmental isolates dated back to 1960s. Five major STs, ST288, ST84, ST54, ST46, and ST289, pointed with arrows, were identified in human cases in Songkhla Province. The STs shown in dark green, red, and blue are identified in humans, animals, and environment sources, respectively, and the sizes represent their frequencies. STs in a dashed circle are clustered together separated from the large group by a double—locus variant (DLV) distance between ST507 and ST164. The number 1 or 2, located on each branch, indicates the genetic distance between two STs either by a single—locus difference or a DLV, respectively.

out of these 29 STs, including all five major STs (84, 54, 46, 51, and 289) identified in that study were also found in patients from our study cohort. Notably, ST51 was also recognized as a major ST frequently found in *B. pseudomallei* strains from Singapore [37]. Given these common STs among *B. pseudomallei* strains in the Malay Peninsula, we hypothesize that most *B. pseudomallei* strains from southern Thailand are likely more closely related to those from Malaysia and Singapore than to those from Northeast Thailand, owing to geographic proximity. This suggests that they might fall within one of the Malay sub-region groups, which currently include only strains from Malaysia and Singapore in the global phylogeny, as previously described [17]. While our current study was based on MLST data, we believe that further investigation, including SNP-based analysis, is warranted to map *B. pseudomallei* strains from southern Thailand onto the global phylogeny. Future studies will include SNP analysis of strains from southern Thailand, Malaysia, Singapore, and Northeast Thailand to provide a more comprehensive understanding of their genetic relationships.

***The presence of B. pseudomallei in natural environment of southern Thailand is mostly associated with animal farms and non-agricultural zones such as mountains, grasslands, waterfalls, streams and surface runoffs.*** Based on the literatures, *B. pseudomallei* is primarily present in rice fields and other agricultural areas in Northeast Thailand. Multiple studies demonstrated the high prevalence of *B. pseudomallei* in soil and water in rice paddies suggesting the vulnerable exposures of rice farmers in Thailand [18,19]. Contamination of *B. pseudomallei* in drinking water linked to infections in rural areas of Northeast Thailand has also been reported [20]. In this study, we conducted a comprehensive collection of soil and water samples from many locations throughout Songkhla Province, the second largest province in the southern region of Thailand. The vegetation found in Songkhla, as well as in other provinces in the South, contrasts with that in the Northeast. Rice cultivation is not predominant covering approximately 3% of the total area of the province, instead, the majority of the agricultural areas, including rubber plantations and fruit farms, are predominantly situated in or around

mountainous regions. Additionally, livestock farms and various horticulture sites are typically established alongside rivers and canals. Despite its low annual melioidosis incidence rate of 2.10 to 3.64 cases per 100,000 population [9] and the association of *B. pseudomallei* presence in the natural environment with non-agricultural areas, the risk of environmental exposure to the infection for residents is considered low compared to the risks faced by rice farmers in the Northeast. Our recent report has indicated that most human cases in Songkhla occurred during the wet seasons, which last up to eight months annually. Although the proportions of *B. pseudomallei*-positive soil and water sources are similar (25% for soil versus 26% for water), several observations lead us to propose that melioidosis in Songkhla Province is predominantly waterborne. Firstly, only 9% of rice fields tested positive for *B. pseudomallei*, indicating a lower prevalence of the bacterium in soil compared to water sources. Secondly, our previous study found that only 21% of melioidosis patients were rice farmers or had occupations involving daily soil/water contact, such as construction workers or park rangers [9]. This suggests that direct soil exposure is not the main route of infection for most patients. Thirdly, we observed a strong association between melioidosis cases and periods of rainfall or flooding, which occur frequently in the region. During these events, floodwaters originating from streams, surface runoffs, and waterfalls, are more likely to be contaminated with *B. pseudomallei*. These water sources are in frequent contact with the population during floods, increasing the likelihood of waterborne transmission. Collectively, these findings support our hypothesis that waterborne routes play a more significant role in the transmission of *B. pseudomallei* in Songkhla Province compared to soil exposure. This reminds us of findings from Townsville, Queensland in Australia, where it was demonstrated that waterborne *B. pseudomallei* from groundwater seeping around Castle Hill potentially facilitated exposure, contributing to clustered cases among residents in that area [38]. In our present study, unfortunately, none of the isolates collected from the environmental sources were genotyped due to funding constraints. As a result, in this study we were unable to determine whether the STs found in patients' isolates are clustered according to the environmental source of infection in the areas they lived. We can only confirm the presence of *B. pseudomallei* in soil or water in most districts of Songkhla Province where patients lived (Table 3).

Lastly, besides developing a regional genetic database of *B. pseudomallei*, the findings of this study underscore the importance of recognizing melioidosis in southern Thailand and emphasize the urgent need for enhanced disease surveillance and management within this region. We believe that our finding could illuminate the contemporary landscape of *B. pseudomallei* infections and their prevalence in the natural environment contributing to the regional risk assessment of melioidosis in Thailand and Southeast Asia.

## Supporting information

**S1 Table. List of *B. pseudomallei* isolates used in this study and their genotypes.** (XLSX)

**S2 Table. List of soil and water sampling sites in Songkhla Province.** (XLSX)

## Acknowledgments

This work is dedicated to the cherished memory of Dr. Richard Alan Finkelstein (1930–2023) whose invaluable contributions, resources, and guidance have left an indelible mark on our field. Special thanks are due to Dr. Michael H. Norris, Dr. Yuta Kinoshita, Aleeza T. Kessler, Bryn E. Tolchinsky, and Sarah Parker for their technical assistances. We acknowledge the

assistance of Ms. Ratana Tongyoi at the Southern Regional Center of Geo-Informatics and Space Technology Prince of Songkla University for her GIS expertise.

## Author Contributions

**Conceptualization:** Apichai Tuanyok.

**Data curation:** Jedsada Kaewrakmuk, Sarunyou Chusri, Apichai Tuanyok.

**Formal analysis:** Jedsada Kaewrakmuk, Sarunyou Chusri, Apichai Tuanyok.

**Funding acquisition:** Kwanjit Duangsonk, Apichai Tuanyok.

**Investigation:** Jedsada Kaewrakmuk, Sarunyou Chusri, Apichai Tuanyok.

**Methodology:** Jedsada Kaewrakmuk, Sarunyou Chusri, Pacharapong Khrongsee, Soontara Kawila, Apichai Tuanyok.

**Project administration:** Jedsada Kaewrakmuk, Vannarat Saechan, Kwanjit Duangsonk, Apichai Tuanyok.

**Resources:** Soontara Kawila, Nutjamee Leesahud, Bongkoch Chiewchanyont, Apichai Tuanyok.

**Supervision:** Hathairat Thananchai, Kwanjit Duangsonk, Apichai Tuanyok.

**Validation:** Apichai Tuanyok.

**Writing – original draft:** Jedsada Kaewrakmuk.

**Writing – review & editing:** Hathairat Thananchai, Kwanjit Duangsonk, Apichai Tuanyok.

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
