## [Decision Letter · Decision Letter 0]

30 Apr 2024

Dear Dr. Tuanyok,

Thank you very much for submitting your manuscript "A molecular epidemiological analysis of Burkholderia pseudomallei in southern Thailand." for consideration at PLOS Neglected Tropical Diseases. As with all papers reviewed by the journal, your manuscript was reviewed by members of the editorial board and by several independent reviewers. In light of the reviews (below this email), we would like to invite the resubmission of a significantly-revised version that takes into account the reviewers' comments. 

The reviewers felt this was a well-performed study with strong results and of public health importance. However, it was noted that complete NGS data is available for these samples, but they have been analysed only by MLST. Given that the NGS data is already available, the samples can be also analysed at the SNP level, which would allow you to answer some of the questions posed by reviewer #1 and would strengthen the conclusions of your paper. For this reason I have recommended a major revision to incorporate whole genome SNP analysis into this study.

We cannot make any decision about publication until we have seen the revised manuscript and your response to the reviewers' comments. Your revised manuscript is also likely to be sent to reviewers for further evaluation.

Sincerely,

Elizabeth M Batty

Academic Editor

Stuart Blacksell

Section Editor

The reviewers felt this was a well-performed study with strong results and of public health importance. However, it was noted that complete NGS data is available for these samples, but they have been analysed only by MLST. Given that the NGS data is already available, the samples can be also analysed at the SNP level, which would allow you to answer some of the questions posed by reviewer #1 and would strengthen the conclusions of your paper. For this reason I have recommended a major revision to incorporate whole genome SNP analysis into this study.

Reviewer's Responses to Questions

**Key Review Criteria Required for Acceptance?**

**Methods**

-Are the objectives of the study clearly articulated with a clear testable hypothesis stated?

-Is the study design appropriate to address the stated objectives?

-Is the population clearly described and appropriate for the hypothesis being tested?

-Is the sample size sufficient to ensure adequate power to address the hypothesis being tested?

-Were correct statistical analysis used to support conclusions?

-Are there concerns about ethical or regulatory requirements being met?

Reviewer #1: Generally: The design of the study is comprehensible and the methods chosen are largely appropriate. However, my main concern is the use of MLST analysis, SNP analyses should be performed for the comparison of related isolates. The necessary data are available, which is why it is surprising that they are only used for the assignment of the ST.

• There is a discrepancy in the numbers see line 38 and line 119.

“A total of 263 clinical isolates retrieved from 156 melioidosis patients” and “In this study, 258 B. pseudomallei isolates from 158 patients were used”

• Line 117: “Multiple B. pseudomallei isolates from different diagnosed specimens were collected from each patient” and Line 240: "To study the strain diversity in individual infection, at least two B. pseudomallei isolates were collected from cultures of available clinical specimens (e.g. blood, sputum, pus, urine) in 59 patients. "

The authors write that they are investigating the genetic diversity of pseudomallei in patient samples. However, it is not clear to me how the isolates were selected. Were all isolates analyzed here, or a few random ones? Please clarify the sentences above. What was the basis on which the strains were selected if not all were analyzed.

• The methods for the detection of B. pseudomallei in the environment sound plausible. Do they correspond to those used in the field or to the consensus guidelines? The authors might provide citations for clarification

• Please, explain the purpose of the different filters for the water filtrate culture.

• Line 183: what enrichment broth was used?

• Line 186: “a) Direct filtrate culture” should be “B)” according to line 172 and Fig.1

as in Line 190: “b) Enriched filtrate culture” should be “C)” according to line 172 and Fig.1

• Please deposit the acquired NGS data to public databases and provide the Accession No. in the manuscript

• Line 214 – 218: In contrast to the other parts of the methods, too little information is contained here. Please provide names of the tools and parameters used for the generation of the assemblies.

• Major: The analysis should not be limited to MLST but should definitely be extended to an SNP analysis.

Reviewer #2: The experimental design appears sound, and the method section is well-articulated, covering most of the essential information. Nevertheless, a few issues require further attention.

Primarily, the authors mentioned "the whole genome sequencing data" in line 205 without providing a data availability statement. Additionally, in line 215, they referred to using various programs on Galaxy for assembly. However, they still need to elaborate on how they determined the optimal results using different programs or pipelines.

Secondly, the manuscript omits the in silico analysis of MLST, a crucial aspect of the method section. It would be beneficial to include these details in the manuscript.

**Results**

-Does the analysis presented match the analysis plan?

-Are the results clearly and completely presented?

-Are the figures (Tables, Images) of sufficient quality for clarity?

Reviewer #1: In my opinion, analysis at ST level is not too informative here, especially since NGS data is available. Even with identical STs, major genetic differences would not be surprising. I recommend an analysis that makes use of the NGS data. A lot of interesting questions come to mind, e.g. How similar are patient isolates with the same ST - between patients and isolates from a single patient? How do patient isolates from the same patient with different STs compare? Are there any special genetic features of isolates with STs that occur more frequently in patients?

• The reader should be introduced to BTFC&YLF genomic groups.

• Line 258: Please rephrase. 2 samples per location would be 416?

• The paragraphs on the isolation of Bp from soil and water are somewhat redundant with the corresponding tables.

• The authors have characterized the soil samples in detail. However, the results remain very descriptive. Here the study might benefit from a more in-depth analysis.

Reviewer #2: The manuscript's data is meticulously organized and presented with utmost clarity, indicating a high proficiency level and an understanding of the subject matter. The authors have demonstrated exceptional attention to detail and have managed to present the information concisely and understandably. This level of organization and presentation is commendable and should be maintained in future work. Overall, the data presented in this manuscript is exemplary and highlights the author's capabilities in the field of study.

**Conclusions**

-Are the conclusions supported by the data presented?

-Are the limitations of analysis clearly described?

-Do the authors discuss how these data can be helpful to advance our understanding of the topic under study?

-Is public health relevance addressed?

Reviewer #1: The relevance of the study for public health is evident.

Still, not only the results but also the conclusions that can be drawn will benefit from an SNP analysis.

Line 407-408: An SNP analysis would substantiate the hypothesis (including strains from southern Thailand, Malaysia, Singapore and, for example, northeast Thailand in the analysis).

Line 433: Although the proportions of B. pseudomallei-positive soil and water sources are similar (25% for soil versus 26% for water), we propose that melioidosis in Songkhla Province is predominantly waterborne.” Please explain why?

Reviewer #2: The data that has been presented in this study provides strong evidence to support the conclusion that has been drawn. The findings have been conveyed through a well-structured discussion that includes a detailed analysis of the data, as well as a thorough examination of the relevant literature. The figures that have been included in the study are not only visually appealing but are also very easy to understand, making it easy for readers to grasp the findings of the study. Overall, this study has been conducted rigorously and systematically, and the results are highly credible and informative.

**Editorial and Data Presentation Modifications?**

Reviewer #1: (No Response)

Reviewer #2: please review the comments for Methods

**Summary and General Comments**

Reviewer #1: In their study "A molecular epidemiological analysis of Burkholderia pseudomallei in southern Thailand." Jedsada Kaewrakmuk and colleagues report the results of their MLST-based typing of patient isolates of the pathogen B. pseudomallei from southern Thailand. The study is complemented by an extensive investigation of the prevalence of the pathogen in the environment. The study is certainly of relevance to policy makers and the melioidosis community, not only in Thailand but also in other endemic areas.

The introduction provides a good detailed overview of the pathogen, the disease and its prevalence and discusses the need and relevance of environmental screening in this region. The latter could be shortened somewhat. In my opinion, the introduction does not cover previous studies enough. E.g. what is the current state of the literature on genetic studies on B. pseudomallei (MLST, or also SNP-based)? The same applies to the environmental prevalence of B. pseudomallei and studies comparing patients and environmental isolates. Finally, the introduction should also address factors that are subsequently investigated in the study (e.g. serogroups, BTFC&YLF genomic groups etc), especially for readers who are not familiar with the field.

Parts of the discussion should already be mentioned in the introduction, e.g. the description of the Finkelstein study. It could then be shortened in the discussion.

The part of the discussion that refers to the STs of the isolates should be shortened to make it easier to follow.

Line 386: “According to the literature review…” The whole paragraph is missing citations.

Line 388-391: It is not entirely clear to me whether the authors suggest that relapse, recurrent infection or reinfection can be distinguished by STs? Here, one would probably prefer to use an analyses with higher resolution (?), because as this study shows, certain STs have been present in the environment for a very long time. If I have misunderstood something, please rephrase.

Reviewer #2: The manuscript is well-written and organized, with a reliable experimental design. However, a data availability statement needs to be included. It's crucial to provide more details on the in-silico analysis of MLST. The findings are well-supported by solid evidence and conveyed through a well-structured discussion. The study has been conducted rigorously and has produced credible and informative results.

PLOS authors have the option to publish the peer review history of their article (what does this mean?). If published, this will include your full peer review and any attached files.

Reviewer #1: No

Reviewer #2: No
---

## [Decision Letter · Decision Letter 1]

7 Aug 2024

Dear Dr. Tuanyok,

We are pleased to inform you that your manuscript 'A molecular epidemiological analysis of Burkholderia pseudomallei in southern Thailand.' has been provisionally accepted for publication in PLOS Neglected Tropical Diseases.

Best regards,

Elizabeth M Batty

Academic Editor

Stuart Blacksell

Section Editor

Reviewer's Responses to Questions

**Key Review Criteria Required for Acceptance?**

**Methods**

-Are the objectives of the study clearly articulated with a clear testable hypothesis stated?

-Is the study design appropriate to address the stated objectives?

-Is the population clearly described and appropriate for the hypothesis being tested?

-Is the sample size sufficient to ensure adequate power to address the hypothesis being tested?

-Were correct statistical analysis used to support conclusions?

-Are there concerns about ethical or regulatory requirements being met?

Reviewer #1: (No Response)

Reviewer #2: The authors have taken the necessary steps to incorporate the feedback I provided.

**Results**

-Does the analysis presented match the analysis plan?

-Are the results clearly and completely presented?

-Are the figures (Tables, Images) of sufficient quality for clarity?

Reviewer #1: (No Response)

Reviewer #2: (No Response)

**Conclusions**

-Are the conclusions supported by the data presented?

-Are the limitations of analysis clearly described?

-Do the authors discuss how these data can be helpful to advance our understanding of the topic under study?

-Is public health relevance addressed?

Reviewer #1: (No Response)

Reviewer #2: (No Response)

**Editorial and Data Presentation Modifications?**

Reviewer #1: (No Response)

Reviewer #2: please review the comments for Methods

**Summary and General Comments**

Reviewer #1: The authors have made significant improvements to the manuscript through their revisions. The amendments to the text have resulted in a more accessible and coherent reading experience, with the introduction providing a clear and effective overview of the subject matter. The incorporation of additional methodological detail and the release of the sequencing data guarantee the reproducibility and further utilization of the data.

A few minor amendments are recommended prior to publication:

• A total of 126 NGS datasets are referenced in the Methods section. I have to admit that I have lost track of the numbers, but this value does not match the numbers mentioned in the text. To ensure clarity and consistency, an explanation of how this number was derived would be beneficial.

• Additionally, in the legend to Fig. 3, arrows are mentioned. They were visible in the initial submission, but are no longer present in the revised version, at least in my version.

• The following sentence in the discussion should be revised as it reads as though it was a response to a reviewer: “We agree that using a higher resolution genotyping technique, such as whole genome SNP analysis, would be more effective in differentiating between relapse and infection. Given that certain STs have been present in the environment for a long time, a comprehensive comparative genomic analysis between clinical and environmental strains sharing the same ST would provide valuable insights into genetic changes, such as gene acquisition or loss, over time.”

It is still my view that a SNP analysis would have been a valuable addition. The effort required would be reasonable (for example, using Snippy) and the benefits substantial. Nevertheless, the study aligns perfectly with the scope of PNTD and will be highly valued by the research community, which is why I recommend its publication.

Reviewer #2: (No Response)

PLOS authors have the option to publish the peer review history of their article (what does this mean?). If published, this will include your full peer review and any attached files.

Reviewer #1: No

Reviewer #2: No

---

## [Editor Report · Acceptance letter]

15 Aug 2024

Dear Dr. Tuanyok,

We are delighted to inform you that your manuscript, "A molecular epidemiological analysis of *Burkholderia pseudomallei* in southern Thailand.," has been formally accepted for publication in PLOS Neglected Tropical Diseases.

Best regards,

Shaden Kamhawi

co-Editor-in-Chief

Paul Brindley

co-Editor-in-Chief
